# Microstructural Considerations of a Multi-Pass Rolled Ti-Nb-Ta-Zr Alloy

**DOI:** 10.3390/ma16083208

**Published:** 2023-04-19

**Authors:** Doina Răducanu, Anna Nocivin, Vasile Dănuț Cojocaru, Nicolae Șerban, Nicoleta Zărnescu-Ivan, Raluca Elena Irimescu, Bogdan Mihai Gălbinașu

**Affiliations:** 1Department of Metallic Materials Processing and Environmental Engineering, University Politehnica of Bucharest, 060042 Bucharest, Romania; doina.raducanu@upb.ro (D.R.); dan.cojocaru@upb.ro (V.D.C.); nicolae.serban@upb.ro (N.Ș.); nicoleta.zarnescu@upb.ro (N.Z.-I.); 2Faculty of Mechanical, Industrial and Maritime Engineering, Ovidius University of Constanta, 900527 Constanța, Romania; anocivin@univ-ovidius.ro; 3Dental Medicine Faculty, University of Medicine and Pharmacy “Carol Davila” Bucharest, 020021 Bucharest, Romania; bogdan.galbinasu@umfcd.ro

**Keywords:** beta-titanium alloys, multi-pass rolling, recrystallization, XRD, SEM

## Abstract

The microstructural characteristic evolution was investigated during thermomechanical processing of Ti-29Nb-9Ta-10Zr (wt %) alloy, which consisted of, in a first stage, in a Multi-Pass Rolling with increasing thickness reduction of 20%, 40%, 60%, 80%, and 90%; in step two, the multi-pass rolled sample with the highest thickness reduction (90%) was subjected to a series of three variants of static short recrystallization and then to a final similar aging. The objective was to evaluate the microstructural features evolution during thermomechanical processing (phase’s nature, morphology, dimensions, and crystallographic characteristics) and to find the optimal heat treatment variant for refinement of the alloy granulation until ultrafine/nanometric level for a promising combination of mechanical properties. The microstructural features were investigated by X-ray diffraction and SEM techniques through which the presence of two phases was recorded: the β-Ti phase and the α″-Ti martensitic phase. The corresponding cell parameters, dimensions of the coherent crystallite and the micro-deformations at the crystalline network level for both recorded phases were determined. The majority β-Ti phase underwent a strong refinement during the Multi-Pass Rolling process until ultrafine/nano grain dimension (about 9.8 nm), with subsequent slow growing during recrystallization and aging treatments, hindered by the presence of sub-micron α″-Ti phase dispersed inside β-Ti grains. An analysis concerning the possible deformation mechanisms was performed.

## 1. Introduction

For hard tissue replacement, titanium and its alloys represent an important metallic biomaterial due to its excellent biocompatibility and high mechanical strength, comparable to that of cobalt alloys or stainless steel, but with low Young modulus that favors the reducing of the well-known “stress shielding effect” [1,2,3]. For permanent bone implants that require high strength coupled with low elastic modulus, the β-type titanium alloys are most appreciated among other two possible types, α- or α/β-type Ti-alloys [4,5,6,7]. Therefore, β-Titanium alloys with non-toxic Nb, Ta, and Zr alloying elements are of maximum interest nowadays [1,2,8,9,10]. The TNTZ (Ti-Nb-Ta-Zr) alloys represent an appreciated selection of β-Ti alloys not only for their non-toxic and biocompatibility character, but also for good β-stabilizing capacity of alloying elements which is required knowing that the single β-phase structure has an easier ability for mechanical processing [11,12]. The β-stabilizing character of niobium has been well known for a long time [9,10]. The same applies for the tantalum [13,14]. Regarding zirconium, even if it is classified as a neutral alloying element, it can favor the stabilization of the β phase in the presence of β stabilizers [15], in this case Nb and Ta. Also, the toughness of titanium alloys can be increased by zirconium, as niobium does [16,17], and the association of both, niobium and zirconium, facilitate a decrease of the elastic modulus [2]; finally, the suppression of the formation of the omega phase assured by zirconium presence [18] represents another important benefit of TNTZ alloys. Thus, referring entirely to the improvement of the mechanical biocompatibility of β-type TNTZ alloys, apart from the above chemical composition, one of the most used and recognized as beneficial methods is the thermomechanical processing, which includes, among others, the grain refinement of the alloy to ultrafine or even nanometer dimensions [3,19,20,21]; this nanometric structure can assure the alloy necessary bio-structural and bio-mechanical properties dictated by particular implant destination/localization and by other mechanical and biochemical demands. A single β-phase structure has an easy ability for mechanical processing, and can achieve a low elastic modulus, but with a moderate mechanical strength [20,22,23,24]; generally, the obtained elastic modulus can be around 55 GPa, which is a satisfactory value comparative to that for cortical bone of about 30–35 GPa [25], but with not-so-high strength, generally not exceeding 550 MPa, which may be insufficient for hip implant applications [14]. For higher strength values, a suitable way may be a structural combination of two phases [26,27]: the majority β phase and a small amount of orthorhombic α″ martensitic phase, depending on the β-phase stability, sufficient to ensure a desirable increase in the mechanical strength of the β phase, but with keeping the low modulus of elasticity through a controllable dispersion, dimension, and quantity of α″ phase [12,28,29,30]. Comparative to hexagonal-α′ martensitic phase or to ω phase, the orthorhombic-α″ martensitic phase has the lowest Young modulus [18], even comparable to that of the β phase [31], but with the possibility of strength enhancing.

Thus, it follows that, by combining the two aspects, i.e., the creation of a structure consisting of β phase with a small/controllable amount of orthorhombic martensitic α″ phase, on the one hand, and the refinement of the structure granulation, on the other hand, an efficient method can result for improving the mechanical biocompatibility of TNTZ alloys. The present study follows this objective and proposes for the experimental program the Ti-29Nb-9Ta-10Zr (wt %) alloy, that is, part of the β-type TNTZ system which has been shown the most appreciated results regarding good biocompatibility and low Young modulus [32,33]. The chemical composition has been selected based on the abovementioned considerations regarding the influence of Nb, Ta, and Zr as alloying elements; thus, to unsure a high stability of β phase, the related literature reports the following necessary intervals: Nb between 20 and 40 wt %, Ta between 7 and 13 wt %, and Zr between 3 and 10 wt % [12,34,35,36]. The selection of the proposed chemical composition also took into account the promising results of already-experimented alloys with close chemical compositions: Ti-29Nb-13Ta-4.6Zr [7], Ti-30Nb-6Zr-5Ta [7,21], Ti-36Nb-2Ta-3Zr-0.3O [22], Ti-29Nb-13Ta-5Zr [37], and Ti–35Nb–3Zr–2Ta [38].

As concerning obtaining an efficient grain refinement combined with a controllable stress-induced martensitic phase formation, in general, one of the severe plastic deformations (SPD) methods can be applied: *multi-pass rolling* (MPR), *accumulative roll bonding* (ARB), *equal-channel angular pressing* (ECAP), and *high-pressure torsion* (HPT) [39,40,41,42]. The desired grain refinement can be achieved through one or more SPD structural mechanisms such as dislocation slip, twinning, and *stress-induced martensite* (SIM) formation [36,43], depending on the stability of the β phase and of the deformation extent [39,40,44]. For the present study, the MPR method has been selected as a part of a complex thermomechanical process proposed for experiments together with a static short recrystallization and aging treatments capable of obtaining ultrafine/nanometric grains with an optimal combination of mechanical properties. Consequently, the aim of the present study is to examine the effects of MPR severe plastic deformation in conjunction with some heat treatments (recrystallization and aging) on the evolution of the Ti-29Nb-9Ta-10Zr alloy microstructure through XRD analysis and SEM visualizations. 

## 2. Materials and Methods

The experimental program (Figure 1) comprises the following stages: the obtaining of the initial sample—S-1, with the proposed chemical composition necessary to perform the multi-pass rolling (MPR) procedure; the applying of the MPR process to the initial stage with increasing deformation degree using a total thickness reduction of 20%, 40%, 60%, 80%, and 90% in order to reduce the grain size step-by-step, until ultrafine/nanometric size, by severe plastic deformation; the applying of a series of short recrystallizations and aging treatments on the highest deformed sample (90%) in order to study the process of microstructure remodeling after a heavy deforming and hardening of the sample; the microstructural analysis of all experimented stages through XRD analysis and SEM visualizations in order to establish the phase nature changing with all dimensional and morphological evolutions.

The proposed alloy for the experimental program has the following chemical composition (wt %): Titanium—52%; Niobium—29%; Tantalum—9%; Zirconium—10%. Considering that the elements amount with β-stabilizing character represents almost one half of the entire composition, it is expected that the structure of the studied alloy can be included in the class of near-beta titanium alloys. This aspect will be analysed by XRD envisaged analysis. The obtaining of the initial sample lot (S-1) of the studied alloy starts from high-purity elemental components and comprises the following stages: the alloy synthesis in a levitation induction melting furnace, FIVE CELES—MP25 (Five’s Group Company, Paris, France), with a nominal power of 25 kW, melting capacity of 30 cm^3^, and using a high vacuum of 10^−4^–10^−5^ mbar. Then, the obtained sample lot in as-cast condition was subjected to a homogenisation treatment at 1223 K (950 °C) with holding time of 6 h, followed by furnace cooling for obtaining a homogeneous structure without casting defects. After that, for refinement of the coarse homogenised microstructure, a plastic deformation by cold rolling was applied with a total thickness reduction of about 60%. Finally, prior to the multi-pass rolling (MPR) procedure, a completely alloy recrystallization was performed at 1123 K (850 °C)/30 min/water quenching (w.q.). The whole above thermo-mechanical processing route applied for obtaining the S-1 structural state has been established based on prior results [45,46,47]. The cold rolling was applied using a Mario di Maio LQR120AS rolling mill (Mario di Maio Inc., Milano, Italy) with a 3 m/min rolling speed and no lubrication; before this procedure, the homogenized alloy was cleaned using an ultrasonic bath at 60 °C in ethylic alcohol. The applied heat treatments were realized in a GERO SR 100 × 500-type oven (Carbolite-Gero Inc., Neuhausen, Germany) under a high vacuum.

The MPR process of the above-established S-1, measuring 20 mm × 20 mm × 60 mm, was applied using the same Mario di Maio LQR120AS rolling mill with a 3 m/min rolling speed and no lubrication. The total reduction degree (ε_tot_) for each of five applied steps was successively increased from 20% to 40%, 60%, 80%, and finally 90%. For each of these variants, the MPR process was performed with a reduction of 10% per pass.

The samples processed by MPR with ε_tot_ = 90% were subjected to a series of three variants of static short recrystallization, R1 (780 °C/5 min/w.q.), R2 (830 °C/5 min/w.q.), and R3 (880 °C/5 min/w.q.), and a final aging-A (400 °C/5 min/w.q.) similar for all recrystallization variants (R1 + A; R2 + A; R3 + A) in order to find the optimal variant for reforming the strong deformed shape of the grain in an equiaxial one by kipping as much as possible the obtained ultrafine/nanometric size of the grains. The same GERO SR 100 × 500-type oven (Carbolite-Gero Inc., Neuhausen, Germany) under a high vacuum as for the homogenization treatment was used for these heat treatments too; the rate of temperature increase was about 15 °C/s. 

The microstructural features evolution was highlighted through X-ray diffraction (XRD) analysis completed with scanning electron microscopy (SEM) analysis. The XRD analysis was performed at room temperature using a Philips PW 3710 diffractometer, with Cu Kα12 radiation source), and with a scanning interval in the range of 30–90° for the 2θ (°) using a step size of 0.02°. The rolling direction was set to be parallel to the direction of X-rays projected onto the sample surface. The main objective of this analysis was to determine the formed phases with their crystallographic characteristics. Therefore, the recorded XRD patterns were simulated and fitted. For the XRD pattern simulation has been used the *MAUD v2.33* software package, by which the phase lattice parameters were calculated. The fitting procedure was performed using the *PeakFit v4.12* software package for determining the position and intensity of each peak, and the peak broadening—FWHM (*Full Width at Half Maximum*). It must be mentioned that, in the fitting procedure, a pseudo-Voigt diffraction line profile was used.

The microstructural analysis was completed by scanning electron microscopy (SEM) visualization using a SEM microscope—TESCANVEGA II—XMU (Tescan Orsay Holding, Brno, Czech Republic). For better observing the evolution of the grain texturing/deformation, the MPR samples were examined mainly in the RD–ND cross section (RD—rolling direction; ND—normal direction).

For preparing the metallographic samples for microscopically observation and the test surfaces for XRD, the following stages were applied: the alloy cutting using a Metkon MICRACUT 200-type machine (Metkon Instruments Inc., Bursa, Turkey) with diamond cutting disks; the fixing of the obtained specimens on a specific epoxy resin of a Buehler Sampl-Kwick type, abraded with 1200-grit SiC paper using a Metkon Digi prep ACCURA machine (Metkon Instruments Inc., Bursa, Turkey); the mechanical polishing of the obtained metallographic samples using 6, 3, and 1 μm polycrystalline diamond suspension first, and 0.03 μm colloidal silica secondly, on a Buehler VibroMet2 machine (Buehler Ltd., Lake Bluff, IL, USA). 

## 3. Results and Discussion

The microstructural characterization of the studied Ti-29Nb-9Ta-10Zr (wt %) alloy was made based on X-ray diffraction analysis and SEM imaging. All resulted data were discussed and compared for evaluating the structural stability of the majority β-solid solution, the grain-refinement capability during the applied MPR process of deformation, and the grain morphology evolution during the short recrystallizations and aging applied afterwards. 

In order to ensure a complete analysis of the microstructural characteristics of the tested samples, it was considered useful to determine the two characteristic temperatures, important for a β-type titanium alloy: the β-transus temperature—T_β_ and the starting temperature of the martensitic transformation—Ms.

Considering the temperature T_β_, it strongly depends on the type of alloying elements used, α-stabilizers or β-stabilizers, since the former increase T_β_ and the others decrease T_β_. Neutral elements have no influence on T_β_ [48]. Consequently, for the determination of T_β_, the established equation from [49] was applied, based on which it results that T_β_ = 813 °C for the currently studied alloy.

Referring to the second characteristic temperature—Ms, it can be calculated depending on the alloy composition using the equation in [50], a special formula that was determined with particular reference to β-Ti alloys with high β stability. By applying this formula, it follows that Ms = −40 °C for the currently studied alloy.

### 3.1. The Phase Analysis Resulted from X-ray Diffraction Investigation

The investigations and analysis by XRD diffraction were made on all experimented samples, beginning with the initial state one (S-1) and continuing with samples obtained after each MPR stage, then after MPR (90%) plus three different variants of recrystallization (R1, R2, and R3), and, finally, after MPR (90%) with three recrystallizations and a similar aging (R1 + A; R2 + A; R3 + A). All obtained XRD patterns were fitted to deconvolute observed cumulative diffraction peaks and to obtain for each constitutive peak its position, intensity, and broadening. Figure 2 and Figure 3 present the recorded XRD patterns; in addition, the Appendix A present the detailed zooms of cumulative diffraction peaks and the Rietveld plots for all examined samples (Appendix A).

For all analyzed samples, the presence of two phases in variable proportions, β-Ti solid solution and α″-Ti martensitic phase, can be observed as a microstructural common denominator. No other secondary phases were recorded. By indexing, it results that the β-Ti phase corresponds to the body-centered cubic (bcc) system—Im-3m space group, and the α″-Ti phase corresponds to the orthorhombic system—Cmcm space group. For both presented phases, the corresponding lattice parameters (a for β phase, and a, b, c for α″ phase) were calculated based on data corresponding to peaks’ position; the obtained data are indicated in Table 1. 

The major diffraction peaks observed for the β-Ti phase are (110) and (211), and for the α″-Ti phase major peaks are (020), (002), (220), and (202). To better highlight these observations, detailed zooms of cumulative diffraction peaks corresponding to 2θ = (37–40°) and 2θ = (68–72°) scattering angles were realized for all samples (Figure 2 and Figure 3). For the magnified interval 2θ = (37–40°) of scattering domain, the sequence order of the cumulative indexed peaks is: α″(020)–β(110)–α″(002); for the second magnified interval 2θ = (68–72°) of scattering domain, the sequence is α″(220)–β(211)–α″(202). 

As concerning the intensities of the major diffraction peaks, those corresponding to β-solid solution are evidently much higher than for the martensitic α″ phase, showing by this the stability of β phase and its majority presence. Another interesting aspect concerns the dominant β-phase peak intensities, the (110) and (211) peaks, observed for all experimented variants: the strongest peak among the β planes in all samples is the (110) peak constantly, followed by the (211) peak. It is well known that the common rolling textures of strong {001}_β_<110>_β_ and weak {211}_β_<110>_β_ for body-centered cubic (bcc) metals are formed by cold rolling [51]; however, the (211) peak becomes, vice versa, greater than (110) for the sample MPR 90% + R1, indicating by this that the texture is changed in this case, a phenomenon which could be correlated with the heating temperature applied for R1 (780 °C), which is lower than the β-transus temperature determined above (813 °C); for the following two recrystallizations, the heating temperatures are higher than β-transus (830 °C and 880 °C, respectively).

The data from Table 1 indicate that the crystallographic cell parameter (a) of the β phase remains essentially constant (~0.330 nm), the applied severe deformation being accommodated either by a twinning process, which is reported to help active the β-Ti bcc grain refinement during the SPD process [34,52,53], and by a SIM process, reported also to occur in some β-Ti alloys [35,54,55]. The SIM process is proved by XRD patterns, but, due to tiny and broad peaks of the recorded orthorhombic-α″ martensitic phase, the proportion of the formed phase is smaller than for the β phase. The same observation and correlation of constantly preserving the β-phase cell parameter (a) at about 0.330 nm corresponds to recrystallized and aged samples also, when the relaxation of the micro-deformations, internal to the β phase, occur. As for the α″ orthorhombic phase, the three corresponding lattice parameters, a, b, and c, have a slight variation: a small decrease during the MPR process, when a crystallographic texture can develop for the samples highly deformed, and a return during recrystallization and aging.

Comparative to other β-Ti alloys with a more active SIM process during severe plastic deformation [34,54,55], the present studied alloy does not record a substantial activity of this process due to very small recorded diffraction peaks of the α″ phase. Reports about correlations between the Nb, Ta, and Zr chemical amounts, and α″ martensitic phase volume fraction show that the last one decreases (or even its formation can be suppressed) if the Nb content is between 23 and 38 wt %, Ta between 4 and 7 wt %, and Zr is higher than 4 wt % [34,52].

For the samples processed by MPR (Figure 2), the intensity of the α″-phase diffraction peaks (very tiny at the beginning) starts to gradually grow, but slowly, remaining of much lower intensity compared to those of the β phase, resulting that the α″ phase formed by the “stress induced martensite” (SIM) process during the severe plastic deformation—MPR remains a secondary and minor phase. Conversely, the main action is proved to be the refinement of the β grains based on constantly decreasing the β-phase peaks intensity, with visible peaks width broadening. For the Figure 3 corresponding to samples processed by MPR followed by a short recrystallization with three different heating temperatures (780 °C, 830 °C, and 880 °C) and similar short holding time (5 min), and a short aging (400 °C/5 min/w.q.), the intensity of the β-phase diffraction peaks gradually returns to higher values, with sharper peaks, denoting a gradual growing of the β grains.

Table 2 indicate the dimensions of the coherent crystallite (D, nm) and the micro-deformations (ε, %) at the crystalline network level corresponding to both β-Ti and α″-Ti phases that were determined using the Williamson–Hall equation [51] for the FWHM parameter. It can be observed that the coherent crystallite dimensions (D, nm) of the β phase decrease constantly with increasing of the deformation degree due to the abovementioned strong refinement process; the diminution is about three times, from 26.92 nm to ultrafine/nanometric size of 9.84 nm. This can be considered the most important result of the present experimental program.

Regarding the micro-deformations (ε, %) corresponding to β-Ti and α″-Ti phases, the values from Table 2 show that those of the β phase gradually decrease with the deformation degree enhancement (until MPR-60%), from 0.28% to 0.14%, due to twinning accumulation inside β grains followed by their splitting for β-grains refinement; after that, the micro-deformations begin to increase slowly (from 0.14% to 0.22%) due to α″-grain gradual formation. The twinning process is also validated by SEM images. On the contrary, the micro-deformation corresponding to the α″ phase gradually increases with the enhancement of the deformation degree because of the predisposition of the α″ crystallites (even in poor quantities), first to forming and then to accommodate between/inside the β grains. After recrystallization and aging, the β grains increase in size while the α″–martensitic grains dissolute in the β phase and diminish in quantity; therefore, the micro-deformations decrease drastically for both phases present, to a minimum of about 0.04%.

### 3.2. The Phase Analysis Resulted from SEM Imaging

The SEM images obtained after analysis using a scanning electron microscope were grouped in two: Figure 4 for the samples processed by MPR with different deformation degree, and Figure 5 for the samples processed by MPR (90%), followed by three variants of short recrystallization and short aging. The SEM images from Figure 4 and Figure 5 correspond to the RD-ND examined cross section and show the characteristic microstructural features for each tested sample.

The first important aspect to emphasize is related to the fact that the SEM images indicate the majority presence of the β phase. By the presence of a high amount of β-stabilizing alloying elements, 48% cumulative for Nb, Ta, and Zr, it was to be expected that the resulted microstructure will be formed of the majority β phase.

Another important aspect relates to the observation that the cold severe rolled microstructure, composed of visible deformed β grains, becomes more and more elongated and textured along the RD direction (Figure 4); also, at high deformation degree, it becomes fragmented due to shear deformation produced by MPR processing. Starting with the first stage of MPR processing, MPR (20%), the β grains have a visible twinning tendency which favors an increasing successive grain refinement [14,20,51]. The sub-micron grains of the α″-Ti phase, formed by SIM process, are dispersed inside β-Ti grains, but they are not visible from the SEM images due to their nanometric dimensions. The visible elongated microstructural features corresponding to the β phase are in agreement with the XRD results already discussed, showing a cumulative deformation mechanism (more active twinning and additionally active SIM transformation). Following the recrystallization (Figure 5), the β-grain dimension begins to increase back constantly with enhancing of the recrystallization temperature. Thus, the gradual morphological rehabilitation of the previously highly deformed and textured nanometric β grains to an equiaxial shape, with gradually increasing sizes to about 108 nm and 186 nm, preserve however the ultrafine character of the final obtained microstructure. This part of the experiments could make it possible to find an optimal final variant of thermo-mechanical processing, with a good balance of necessary mechanical properties.

As concerning the α″-grain size (Table 2), no significant dimensional variations are observed as in the case of the β phase. However, an interesting variation during the MPR process can be noted, from 7.32 nm (for MPR 20%) to a smaller one 6.84 nm (for MPR 40%), and again to larger dimensions (9.12 nm for MPR 90%). The reason for such variable evolution may be that the α″ grains decrease during MPR, due to an inevitable crushing process (achieving 6.84 nm for the MPR-40%); concomitantly, new α″ grains are forming by SIM process in the following MPR stage, with again increased dimension (8.43 nm for MPR-60%), which will be also crushed subsequently; so, an alternative forming-crushing process occurs with variable α″-grains dimensions, but not significantly.

After recrystallization, the grains of the α″ phase preserve some constancy of their values, but with a strong decrease in the amount of phase, from 10.65% to 2.13%, due to their dissolution process in the β phase. Compared to similar variants of recrystallization and aging, the SEM image corresponding to the [MPR-90% + R1 + A] stage (Figure 5d) shows visibly thicker limits of the β phase due to a higher amount of secondary α″ phase present here (6.34% versus 2.27% or 2.06%), because the heating temperature of 780 °C is lower than β-transus (813 °C) and allows a somewhat more pronounced precipitation of the α″ secondary phase, compared to similar treatments but with higher temperatures than β-transus (830 °C and 880 °C, respectively).

Thus, a distinction should be made between the α″ phase that forms following the SIM transformation by applying a severe plastic deformation, and the same α″ phase that once dissolves into the β phase upon recrystallization and can then precipitate through aging, but not consistently, because the treatment is of short duration (5 min).

In the end, the experimental validation of the microstructural features and other detailed aspects concerning the β-phase stability and its accented refinement process by combining MPR with recrystallization/aging until nanometric dimensions is obtained mainly by XRD data acquisition and processing, and by some relevant SEM imaging. The refinement process of the β phase until nanometric grain dimensions through the experimented thermomechanical process permits the selection of a suitable processing variant function of possible particular aspects of a real application/situation.

## 4. Conclusions

A β-type Ti-based alloy was studied concerning the β-phase stability evolution and the microstructural and crystallographic features evolution when subjected to severe plastic deformation, a MPR processing in this case, and a subsequent combination of short recrystallization and aging.

Using XRD analysis and SEM imaging, the β-phase stability and the ability of α″-martensitic phase formation through stress-induced martensitic transformation were evaluated.

For both recorded phases, the β-Ti phase and α″ martensitic phase, the crystallographic cell parameters, the coherent crystallite dimensions, the phase quantities and the micro-deformations at the crystalline network level were determined by analyzing and processing the data obtained through X-ray diffraction. For verifying and validating the XRD results, the relevant SEM images corresponding to each experimented stage were added.

As a consequence, based on the provided analysis, it has been established/highlighted the following: a good β-phase stability during the entire thermo-mechanical proposed process; the deformation mechanism during MPR process, mainly by twinning and seconded by SIM transformation; the pronounced refinement of the β phase until nanometric grain dimensions. Further studies need to clarify more exactly the involvement, intensity and sequence of the plastic deformation mechanisms that take place in the alloy microstructure by studying the evolution of the mechanical properties.

However, at this stage, the present study provides useful data to start the design of an optimal processing route for the Ti29Nb-9Ta-10Zr alloy, with a good selection of possible ultrafine/nanometric grains capable of suitable mechanical properties.

## Figures and Tables

**Figure 1 materials-16-03208-f001:**
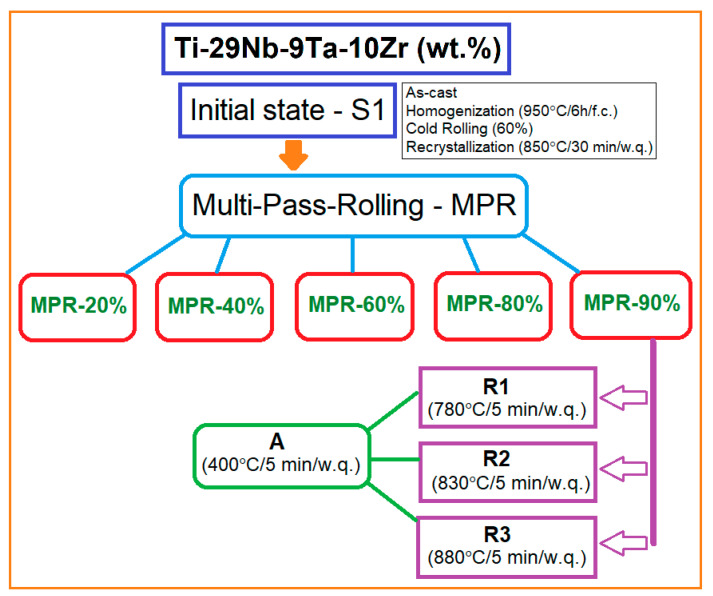
The experimental thermomechanical processing schema of the studied Ti-29Nb-9Ta-10Zr (wt %) alloy.

**Figure 2 materials-16-03208-f002:**
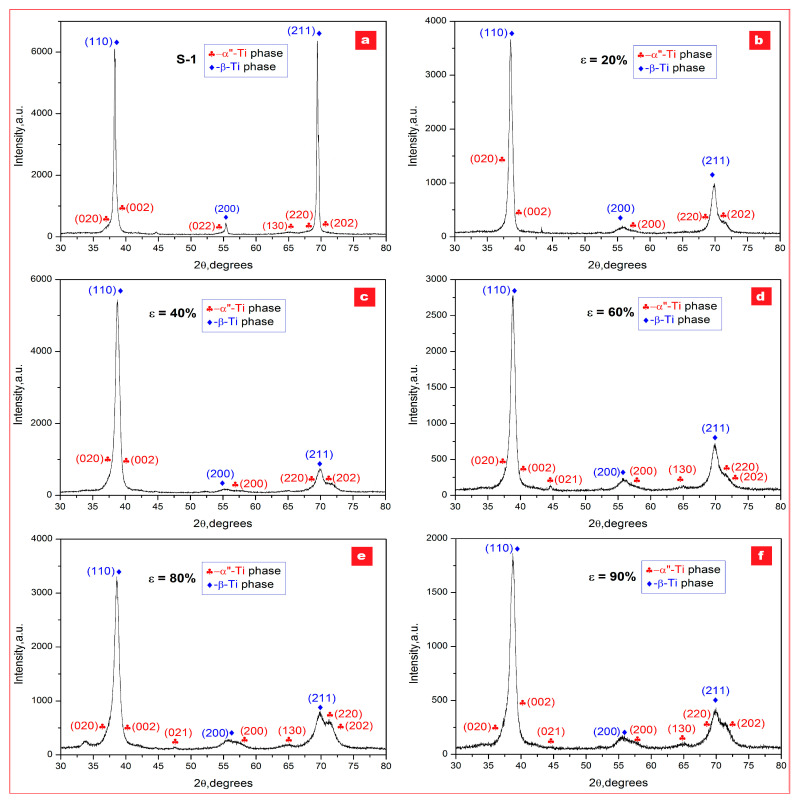
XRD patterns of the Ti29Nb-9Ta-10Zr alloy corresponding to following stages: (**a**) initial state—S-1; (**b**) MPR processed with ε = 20%; (**c**) MPR processed with ε = 40%; (**d**) MPR processed with ε = 60%; (**e**) MPR processed with ε = 80%; (**f**) MPR processed with ε = 90%.

**Figure 3 materials-16-03208-f003:**
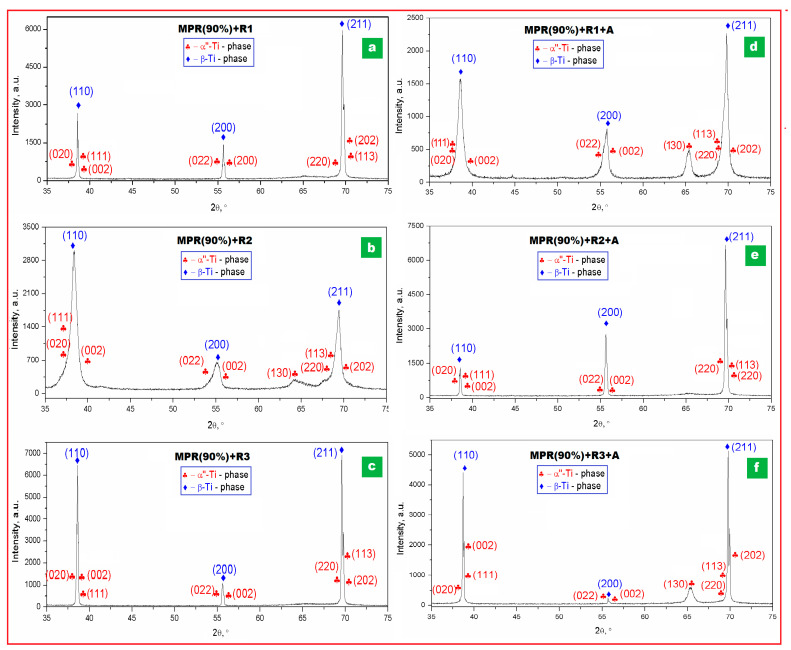
XRD patterns of the Ti29Nb-9Ta-10Zr alloy corresponding to following stages: (**a**) MPR (90%) + R1 (780 °C/5 min/w.q.); (**b**) MPR (90%) + R2 (830 °C/5 min/w.q.); (**c**) MPR (90%) + R3 (880 °C/5 min/w.q.); (**d**) MPR (90%) + R1 + A (400 °C/5 min/w.q.); (**e**) MPR (90%) + R2 + A (400 °C/5 min/w.q.); (**f**) MPR (90%) + R3 + A (400 °C/5 min/w.q.).

**Figure 4 materials-16-03208-f004:**
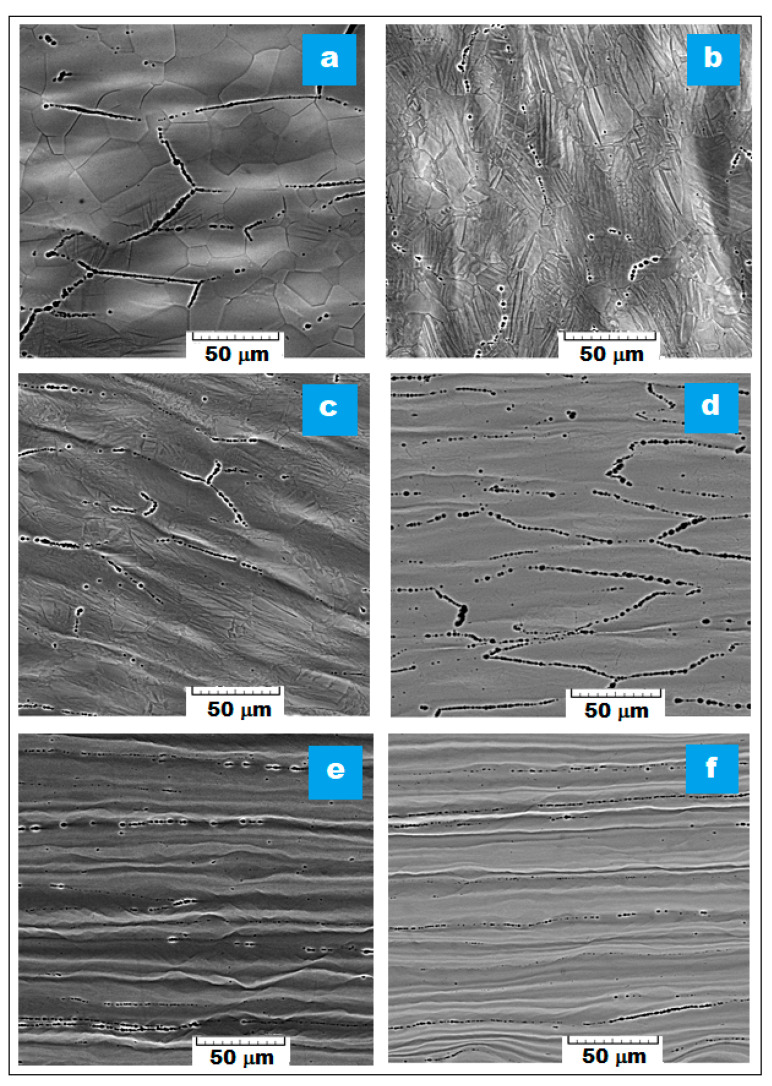
SEM images of the Ti29Nb-9Ta-10Zr alloy corresponding to the following stages: (**a**) initial state—S-1; (**b**) MPR processed with ε = 20%; (**c**) MPR processed with ε = 40%; (**d**) MPR processed with ε = 60%; (**e**) MPR processed with ε = 80%; (**f**) MPR processed with ε = 90%.

**Figure 5 materials-16-03208-f005:**
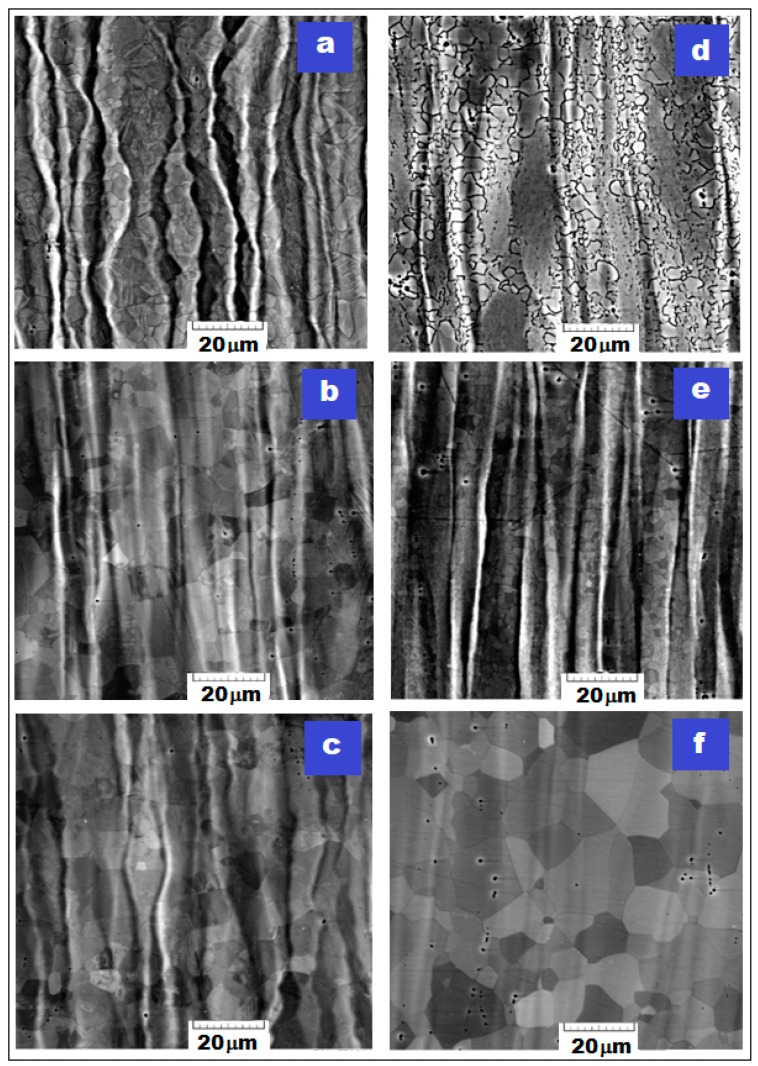
XRD patterns of the Ti29Nb-9Ta-10Zr alloy corresponding to the following stages: (**a**) MPR (90%) + R1 (780 °C/5 min/w.q.); (**b**) MPR (90%) + R2 (830 °C/5 min/w.q.); (**c**) MPR (90%) + R3 (880 °C/5 min/w.q.); (**d**) MPR (90%) + R1 + A (400 °C/5 min/w.q.); (**e**) MPR (90%) + R2 + A (400 °C/5 min/w.q.); (**f**) MPR (90%) + R3 + A (400 °C/5 min/w.q.).

**Table 1 materials-16-03208-t001:** Average crystalline network parameters corresponding to β-Ti and α″-Ti phases identified for all analyzed stages of the studied alloy (MPR—multi-pass rolled sample; R—recrystallized sample; A—aged sample).

Structural State	β-Ti Phase	α″-Ti Phase		
	a [nm]	a [nm]	b [nm]	c [nm]
S-1: Initial Sample	0.330	0.329	0.478	0.462
MPR (εtot = 20%)	0.330	0.323	0.474	0.462
MPR (εtot = 40%)	0.330	0.323	0.472	0.462
MPR (εtot = 60%)	0.330	0.321	0.470	0.460
MPR (εtot = 80%)	0.330	0.322	0.472	0.462
MPR (εtot = 90%)	0.329	0.321	0.471	0.461
MPR (εtot = 90%) + R1	0.331	0.330	0.476	0.466
MPR (εtot = 90%) + R2	0.331	0.333	0.479	0.469
MPR (εtot = 90%) + R3	0.330	0.326	0.476	0.448
MPR (εtot = 90%) + R1 + A	0.330	0.326	0.477	0.469
MPR (ε_tot_ = 90%) + R2 + A	0.331	0.331	0.475	0.465
MPR (ε_tot_ = 90%) + R3 + A	0.330	0.333	0.478	0.464

**Table 2 materials-16-03208-t002:** The average dimensions of the coherent crystallite (D, nm) and micro-deformations (ε, %) corresponding to β-Ti and α″-Ti phases identified for all analyzed stages of the studied alloy (MPR—multi-pass rolled sample; R—recrystallized sample).

Structural State	β-Ti Phase			α″-Ti Phase		
	ε [%]	D [nm]	Phase’s %	ε [%]	D [nm]	Phase’s %
S-1: Initial Sample	0.05	26.92	99.52 ± 0.61%	0.06	-	0.48 ± 0.12%
MPR (εtot = 20%)	0.28	24.16	97.79 ± 1.17%	0.22	7.32	2.21 ± 0.68%
MPR (εtot = 40%)	0.26	11.72	94.19 ± 1.84%	0.28	6.84	5.81 ± 1.09%
MPR (εtot = 60%)	0.14	11.48	91.38 ± 1.31%	0.35	8.43	8.62 ± 1.01%
MPR (εtot = 80%)	0.18	10.19	89.67 ± 1.27%	0.89	7.65	10.33 ± 1.89%
MPR (εtot = 90%)	0.22	9.84	89.35 ± 1.48%	0.92	9.12	10.65 ± 1.04%
MPR (εtot = 90%) + R1	0.36	108.22	95.54 ± 0.85%	0.44	9.33	4.46 ± 0.66%
MPR (εtot = 90%) + R2	0.39	112.63	97.15 ± 0.94%	0.42	9.64	2.85 ± 0.38%
MPR (εtot = 90%) + R3	0.40	138.65	97.87 ± 1.25%	0.41	10.08	2.13 ± 0.07%
MPR (εtot = 90%) + R1 + A	0.04	138.82	93.66 ± 1.34%	0.06	9.31	6.34 ± 0.54%
MPR (εtot = 90%) + R2 + A	0.07	156.76	97.73 ± 0.96%	0.04	7.83	2.27 ± 0.11%
MPR (εtot = 90%) + R3 + A	0.09	186.33	97.94 ± 0.47%	0.04	8.24	2.06 ± 0.09%

## Data Availability

The data presented in this study are available on request from the corresponding author.

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
