# Peer review of "Microstructural Considerations of a Multi-Pass Rolled Ti-Nb-Ta-Zr Alloy"

_materials, 2023, doi:10.3390/ma16083208_

Round 1

Reviewer 1 Report

Manuscript is devoted to X-ray diffraction analysis of Ti-based alloys under mechanical treatment. Since the XRD is of crucial importance for this manuscript one can expect accurate analysis of XRD data. Nevertheless, I found that the XRD data were treated

According to experimental part, line 134, authors used ‘Philips PW 3710 diffractometer, with Cu k-alpha radiation source’, no information about monochromatization was given, thus I guess that Cu Kα12 radiation was applies. The Cu Kα12 radiation contains a doublet of two peaks with the wavelengths 1.540598 and 1.544426Å and relative intensities I(Cu Kα1): I(Cu Kα2) = 2:1. This is common situation for diffractometers equipped by Ni Kβ filter or graphite monochromator. As a result of this doublet nature of Cu Kα12 radiation XRD peaks on patterns also have a doublet profile. The wavelengths are quite close to each other, so at small angles peaks look like singlets, while at high angles peaks become well separated. This effect is clearly visible on XRD patterns presented in this manuscript. Unfortunately, authors treated all XRD patterns as obtained with singlet Cu Kα1. This resulted in wrong phase analysis, wrong lattice parameters, crystallite size and micro-deformation values. For instance, lets consider Fig. 12 (see attached file for details):

Experimental XRD pattern contains two well separated peaks at 69.75 and 69.95º. The Bragg equation is 2d sin θ = λ

Thus, for Cu Kα1, λ1 = 1.5406Å and for the first peak one calculates the inter-planar distance d = Ѕ λ1/ sin θ1 = Ѕ 1.5406 / sin (69.75/2) = 1.347 Å which corresponds to (211) plane of beta-Ti. At the same time, for the second peak within the doublet for Cu Kα2, λ2 = 1.5444Å:

one calculates d = Ѕ λ2/ sin θ2 = Ѕ 1.5444 / sin (69.95/2) = 1.347 Å – the same inter-planar distance. This is additionally confirmed by relative intensities of these two peaks – 2:1. To support this statement I have simulated theoretical XRD patterns of beta-Ti for monochromatic Cu Kα1 and doublet Cu Kα12 radiation and compared it experimental one. It is clearly visible, that both position and intensities of experimental peaks perfectly agree with calculated for Cu Kα12 radiation (see attached file).

Therefore, the second peak correspond to the reflection from Cu Kα2 on (211) plane of beta-Ti , but not to the reflection of Cu Kα1 from α’’-titan plane. The same explanation is for all other peaks on XRD patterns of all samples.

Thus, I have not see a clear signs of α’’-titan phase in XRD data. Please give references on ICDD PDF-2 cards which were used for phase analysis.

All XRD patterns should be re-refined in a specific XRD software (MAUD, JANA2006, Topas, Fullprof, GSAS-II, etc) taking into account a double nature of X-ray radiation. Results of XRD refinement in a form of Rietveld-plot for reppresentative samples should be placed in main text, while major part of XRD patterns should be transferred to supporting information.

Additionally, there are several minor flaws:

1. line 134, ‘with Cu k-alpha radiation source (λ = 0.15406 nm, 40 kV and 30 mA) ’ should be ‘with Cu Kα12 radiation (mean λ = 0.154187, 40 kV and 30 mA)’. please specify monochromatization manner (if so).

2. line 139 and elsewhere ‘XRD spectra ’ better is ‘XRD pattern’

3. Table 1, Table 2. calculated lattice parameters, crystallite size, micro-deformations and phase contents should be presented with standard uncertain values.

Reviewer 2 Report

The subject of research presented in the paper is an XRD analysis of a multi-pass rolled Ti-Nb-Ta-Zr alloy.  The authors of the work researched the microstructural characteristic evolution was investigated during thermomechanical  processing of Ti-29Nb-9Ta-10Zr (wt.%) alloy which consisted of, in a first stage, in a Multi-Pass Rolling with increasing thickness reduction of 20%, 40%, 60%, 80% and 90%; in step two, the multi-pass rolled sample with the highest thickness reduction (90%)  and then similar aging. The objective was toevaluate the microstructural features evolution during thermomechanical processing (phase’s nature, morphology, dimensions and crystallographic characteristics) and to find the optimal heat treatment variant for refinement the alloy granulation until ultra-fine/nano-metric level for promising combination of mechanical properties.

The first part of the publication was devoted to a literature review for the subject selected by the authors. However, this part has been insufficiently described. As a reviewer, I think you should go more into the topic of alloying and rolling. There isn't link to research on rolling of this type of alloys.

In the materials and methods part of the publication, I don't like the designation of studies (A), (B) and (C) (lines 79-89). It would be better to should either be bulleted or removed.

The next part deals with the results and discussion, in which the X-ray diffraction and  microstructural analysis was completed by a scanning electron microscopy.

The entire publication must be formatted.An introduction should be added, and better bulleting of research should be ensured. Tables and figures should be placed next to the text in which they occur.

Round 2

Reviewer 1 Report

Revised manuscript underwent only few corrections of minor flaws.

The major issues are still not resolved. Namely, I guess that the reported peaks of alpha''-Ti phase are just measurement artifacts which originate from the presence of Cu Ka2 peak in X-ray source spectrum.

Authors stated in their response 'Cu Kα12 radiation was applied. No monochromator was used. The peaks were stripped, and fluorescence radiation of the sample was eliminated.' Does it mean application of so-called 'Ka2-stripping' procedure? This routine is not recommended for futher Rietveld analysis because of data distortion. If this procedure was applied I kindly request authors to provide the comparision of original as-measured PXRD pattern and pattern after Ka2-stripping (e.g. for sample on Fig. 12 or Fig. 13) for review.

Since authors reported Rietveld refinement using MAUD software, please provide the Rietveld plots instead of PXRD profiles with best-fit curves obtained in PeakFit v4.12 software.

Manuscript is over-saturated by the similar figures with minimal additional information. Most of these figures should be transfered to Supporting Information.

'

Reviewer 2 Report

The authors of the work entitled "Microstructural Considerations of a Multi-Pass Rolled Ti-Nb-Ta-Zr Alloy" reserch the microstructural analysis stages through a XRD analysis and SEM visualizations in order to establish the phase nature changing with all dimensional and morphological evolutions.

In my opinion, the subject of the paper is interesting and fits well within the scope of the Materials journal. The authors have corrected the errors, resulting in the work gaining quality.
